# Rapid Whole Genome Sequencing in Critically Ill Neonates Enables Precision Medicine Pipeline

**DOI:** 10.3390/jpm12111924

**Published:** 2022-11-18

**Authors:** Makenzie Beaman, Kimberley Fisher, Marie McDonald, Queenie K. G. Tan, David Jackson, Benjamin T. Cocanougher, Andrew P. Landstrom, Charlotte A. Hobbs, Michael Cotten, Jennifer L. Cohen

**Affiliations:** 1Duke University Medical Scientist Training Program, Durham, NC 27710, USA; 2Duke University Department of Pediatrics, Division of Medical Genetics, Durham, NC 27710, USA; 3Duke University Department of Pediatrics, Division of Neonatology, Durham, NC 27710, USA; 4Duke University Department of Pediatrics, Division of Cardiology, Durham, NC 27710, USA; 5Rady Children’s Institute for Genomic Medicine, Rady Children’s Hospital, San Diego, CA 92123, USA

**Keywords:** genome sequencing, neonatology, rapid diagnosis, rapid whole genome sequencing, precision medicine, medical genetics

## Abstract

Rapid genome sequencing in critically ill infants is increasingly identified as a crucial test for providing targeted and informed patient care. We report the outcomes of a pilot study wherein eight critically ill neonates received rapid whole genome sequencing with parental samples in an effort to establish a prompt diagnosis. Our pilot study resulted in a 37.5% diagnostic rate by whole genome sequencing alone and an overall 50% diagnostic rate for the cohort. We describe how the diagnoses led to identification of additional affected relatives and a change in management, the limitations of rapid genome sequencing, and some of the challenges with sample collection. Alongside this pilot study, our site simultaneously established a research protocol pipeline that will allow us to conduct research-based genomic testing in the cases for which a diagnosis was not reached by rapid genome sequencing or other available clinical testing. Here we describe the benefits, limitations, challenges, and potential for rapid whole genome sequencing to be incorporated into routine clinical evaluation in the neonatal period.

## 1. Introduction

As the field of genomic medicine progresses toward more precise diagnostics and therapeutic interventions, it is imperative that we ensure these advancements are quickly and successfully reaching the patients. Numerous studies have shown the benefit of rapid genome sequencing when utilized among a population of critically ill neonates [1,2,3,4,5] and children, [6] and in particular, an improved diagnostic capacity when the analysis of these patients’ DNA included parental samples [3]. The benefits include an economic advantage for the health system [2,4,7,8] and a medical advantage by providing more precise and rapid care to these neonates. Given the required cost and infrastructure needed to complete rapid genome sequencing, the resource is not yet widely available across the USA [9], but efforts by groups such as Rady Children’s through the Vermont Oxford Network Biorepository have helped open this opportunity to additional institutions by partnering with outside hospitals as sites to host pilot studies. Here, we describe the experience of our Duke University Health System site, the outcomes from our pilot study, and our creation of a parallel biorepository to allow for additional research in the cases where rapid genome sequencing did not reveal a diagnostic result.

## 2. Materials and Methods

We report on our experience with a pilot study for rapid whole genome sequencing in the neonatal intensive care unit at Duke University. The pilot was a collaboration between Medical Genetics providers (physicians and genetic counselors), intensivists from the Neonatal, Pediatric, and Pediatric Cardiac Intensive Care Units, clinical research coordinators, laboratory directors, and genomics researchers at Duke University in tandem with and as a site for the Rady Children’s Institute for Genomic Medicine and the Vermont Oxford Rady Children’s Genomic Network. The Duke Institutional Review Board approved the study with a determination of reliance on WCG as the central IRB of record. All patients and families consented through an informed consent process prior to participation in the study.

Prior to the initiation of the study, we completed a comprehensive review of the literature to identify the appropriate inclusion and exclusion criteria for our site (Duke University Hospital), to both maximize our diagnostic likelihood and the utility of this limited resource: rapid genome sequencing analysis conducted as a trio (with parental samples) at no cost to the family or the medical institution. Our site prioritized children with multisystemic and/or severe disease, including multiple congenital anomalies, intractable seizures, metabolic crisis, or cardiac disease. To identify conditions that were most likely to be genetic in nature, we excluded any case thought to be due to isolated sepsis, prematurity, unconjugated hyperbilirubinemia, transient neonatal tachypnea, hypoxic ischemic encephalopathy, trauma, or meconium aspiration. Infants were required to be within one week of admission or within one week of an acute decompensation of an underlying condition (Figure 1).

We educated the relevant parties by presenting our study and inclusion/exclusion criteria at our institution’s Neonatology Grand Rounds as well as the Joint Perinatal–Pediatrics conference, attended by neonatologists, maternal-fetal medicine specialists, medical geneticists, genetic counselors, and pediatric surgeons. We surveyed the prenatal complex cases presented at this conference monthly to identify potential candidates for enrollment. Two of our cases were identified prenatally via this mechanism.

When an infant with features concerning for a genetic condition was delivered at Duke, or transferred to Duke early in life, the intensivist team placed a consult to the Medical Genetics team (Figure 2). The genetics on-call provider evaluated the patient and determined eligibility for the rapid genome sequencing study. In parallel, any clinically available first-line genetic tests that could aid in diagnosis were ordered. Other subspecialty providers involved in a patient’s care were reminded to order all testing that would be standard of care for that patient; in this way, standard of care medical evaluation and testing continued to be conducted without delay, regardless of a patient’s participation in the study. The only testing that was not pursued if a family decided to enroll in this study was rapid clinical exome sequencing, since this is a limited resource and was perceived as redundant and potentially inferior to the clinical-grade rapid genome sequencing being performed through this research study. Following eligibility determination, the genetics team introduced the study to the family. If interested, our dedicated neonatal clinical research team (Neonatal–Perinatal Research Unit) consented the proband and parents to the study and collected blood samples. The clinical research team coordinated blood draws for all consenting subjects and sent samples via overnight shipping to the sequencing lab at the Rady Children’s Institute for Genomic Medicine. The research team at Duke, consisting of a clinical geneticist (site Principal Investigator (PI)), an MD/PhD student, and the Neonatal–Perinatal Research Unit clinical research coordinator, input clinical information into Rady Children’s online portal. This information included demographic and identifying information for subjects, as well as all relevant clinical notes and family history. The Rady Children’s Clinical Laboratory Improvement Amendments (CLIA)-certified genomic lab performed sequencing and implemented phenotype-prioritized variant calling via clinical natural language processing, as previously described [10]. In the case of presumptive findings, a preliminary result was called out to the site PI by a Rady Children’s laboratory director, while orthogonal sequencing confirmation was underway. The results were reported to the primary clinical team and family by a geneticist (either site PI or geneticist on-call) and a genetic counselor.

If the child’s status rapidly declined or their result was inconclusive, the family was approached about further research options. In collaboration with the sequencing study team, a research team consisting of a pediatric physician–scientist, an MD/PhD student, and genomics researchers consented interested probands and parents to a biorepository at Duke. In these cases, whole blood (with specialized tubes for the stabilization of DNA and RNA) was collected, and peripheral blood mononuclear cells were isolated, cultured, and viably frozen. Ongoing collaborative research studies at our institution enable the ability to perform several cutting-edge assays on these samples: long-read DNA sequencing to detect variants missed by whole genome sequencing, such as certain structural variants; RNA sequencing for the evaluation of noncoding variants, alternative splicing, or allele-specific expression leading to disease; and de-differentiation of patient peripheral blood mononuclear cells into induced pluripotent stem cells for modeling of cell phenotypes in inaccessible tissue (such as brain and heart).

## 3. Results

### 3.1. Demographics

Our study was conducted from July 2021 through July 2022. Our team’s consenting algorithm (Figure 1) prioritized children with multisystemic disease. Table 1 demonstrates the demographics and clinical indications of the enrolled probands. Seven probands (87.5%) were enrolled in the neonatal period, while admitted in the neonatal intensive care unit or pediatric cardiac intensive care unit. The median age of these probands at the time of enrollment was 5 days (range 1–10 days). An eighth proband was enrolled from the pediatric intensive care unit at 6 months of age, during an acute decompensation from anomalies present since birth. Our study team collected data on parental ancestry for each family. Self-reported ancestry was critical to the diagnosis in patient 7, who harbored a homozygous pathogenic variant known to be prevalent in Native Americans with the diagnosed condition.

### 3.2. Genetic Results

A conclusive genetic diagnosis was reached in four of eight cases (50%) (Table 2). In three of these cases, the diagnosis was made via rapid genome sequencing with trio analysis: *CAPN15*-related oculogastrointestinal neurodevelopmental syndrome, *TTN*-related disorder, and *STAC3*-related myopathy. In the fourth, a diagnosis of mosaic monosomy was not identified on genome sequencing from whole blood but was instead confirmed with fluorescence in situ hybridization conducted on a buccal sample and performed at the Duke Molecular Pathology, Genetics, and Genomics (MPGG) clinical laboratory. Changes in management were implemented in two cases. For patient 1, familial cascade testing and cardiac surveillance were pursued due to a pathogenic, potentially lethal arrhythmia variant identified incidentally. For patient 7, anesthesia precautions were undertaken to prevent malignant hyperthermia risk associated with the patient’s diagnosed condition. A suggestive variant was found in one individual; any possible confirmatory testing was unable to be completed, however, due to the death of the proband and the family’s decision to not pursue an autopsy. The results were non-diagnostic in three families (37.5%)—all these families consented to further research to investigate potential etiologies that were not identified on rapid genome sequencing.

The average time to result in our entire cohort was 10.6 days. Cases with a positive result were reported faster: the average time to a positive result was 7.3 days, while inconclusive or non-diagnostic results were reported at an average of 14 days.

Four probands in our cohort passed away over the course of the study (50%). Of the patients who received a confirmed genetic diagnosis, three are deceased (75%); of the patients who have not received a confirmed diagnosis, one is deceased (25%). Two probands were deceased prior to the return of their rapid genome result, and two probands were deceased after the result was returned.

## 4. Discussion

While rapid whole genome sequencing is certainly promising for the diagnosis of genetic conditions, certain limitations prevent its widespread clinical adoption. We encountered some of these barriers in our own cohort. We prioritized cases where both parents were available due to the ability to interrogate variant inheritance through trio genome interpretation [3]. Even with this requirement, we faced challenges collecting blood samples for both parents in a timely manner in several cases. Because Duke is a tertiary referral center, patients travel from around the state and region to receive care. Several of our neonates were transferred from regional hospitals many hours away, while the mother remained hospitalized at the birthing hospital. In other cases, a child remained in the NICU while parents returned home to care for other children. In one case, a father tested positive for a respiratory illness and was unable to come to the hospital for in-person consent and sample collection for several days, delaying sequencing and interpretation. Our criterion to include both parents in analyses did not prevent the inclusion of any eligible patients—no nominated probands in our cohort were excluded because of the inaccessibility of one or both parents. In each case where parental sample was delayed, samples were sent for all available family members at the time of allocation, and then remaining samples were processed as soon as they became available. In one case, this resulted in the processing of one family as a duo initially, and a reanalysis as a trio once a maternal sample was obtained. Our site required in-person consent and blood collection at the hospital; the implementation of electronic consents and mobile phlebotomists may alleviate these barriers in other studies.

In cases where the rapid genome is nondiagnostic, it is important to collect additional samples, including blood, biopsy, and/or surgical tissue specimens, for downstream analysis. However, the critically ill status of our enrolled subjects heightened the risk of routine blood draw or surgical procedures. In several cases, coagulopathies or massive transfusion protocols prevented us from collecting additional samples for confirmatory clinical or research testing. Technological advancements allowing testing on noninvasive specimens will be critical to diagnosis in these unstable patients. Postmortem tissue samples from autopsy provide a valuable source of information for infants who pass before a diagnosis is made, though many parents are understandably hesitant to consent to extensive testing after the traumatic death of their infant.

Diagnosis via rapid whole genome sequencing requires that a variant is present in the germline or in leukocytes isolated from whole blood. Patient 2 had variable skin lesions raising suspicion for mosaic involvement. The low-level variant frequency in blood was below the limit of detection for copy number variants for the Rady Children’s computational pipeline but was detected on a clinically obtained chromosomal microarray from whole blood. The diagnosis was confirmed via interphase fluorescence in situ hybridization on a buccal swab of the affected mucosal tissue, where the pathogenic deletion was present in a slightly higher proportion. There is an increasing appreciation for somatic mosaicism as a mechanism of genetic disease; this should always be a consideration when sequencing of blood is negative or inconclusive. The physical exam is critical to determining which cases will benefit the most from whole genome sequencing of blood versus other tissue; an exam suggestive of mosaic involvement should prompt additional consideration of the collection of affected tissue. We are evaluating the possibility of mosaicism in patient 4 in our cohort and have obtained surgical tissue from the affected colon to perform additional genomic analyses.

Rapid genome sequencing relies on phenotype-driven algorithmic prioritization of variants to reduce the manual analysis time and burden. Because of this, patients with nonclassical presentations of disease are easily missed. Patient 1’s case highlights a novel phenotype association with a known disease gene, *CAPN15.* Though *CAPN15*-related disease had been previously described as an autosomal recessive condition, its constellation of features did not fully explain the patient’s presentation. Her sacral and cardiac anomalies overlapped with previously described individuals harboring biallelic variants in *CAPN15*, but no cases had exhibited severe structural brain disease, a major feature of her presentation. Of note, one of her *CAPN15* variants was initially classified as a variant of uncertain significance. Patient 1 had an older sister who had an identical phenotype but no genetic diagnosis to date—she was known to have a chromosomal deletion (which involves the *CAPN15* locus) but had not had exome or genome sequencing. We pursued targeted testing of the opposite allele of the *CAPN15* gene in the older sister and found the same missense variant present in her. Segregation of the putative variant in the two affected individuals in the family upgraded the variant to likely pathogenic and suggested a phenotype expansion of the previously described *CAPN15*-related disorder to include the brain malformations identified in both sisters. The sequencing lab at Rady Children’s has a high threshold for reporting variants in genes without currently established human gene-disease association, requiring compelling inheritance patterns, multiple lines of pathogenicity prediction, and animal model or research evidence to call a variant in a novel disease gene. To keep up with the ever-expanding knowledge base surrounding novel disease genes and genes of uncertain significance, regular reanalysis of sequencing data should be conducted as new information is gleaned.

## 5. Conclusions

We reported a 37.5% diagnostic yield via rapid whole genome sequencing in our pilot cohort, consistent with previous large-scale reports of rapid genome utility via the Rady Children’s analysis pipeline [3,4,5]. An additional suggestive diagnosis was made in our cohort but could not be confirmed before the demise of the infant. One diagnosis was missed via whole genome sequencing but was made with peripheral blood chromosomal microarray and buccal swab interphase fluorescence in situ hybridization, highlighting the continued importance of traditional first line genetic tests in combination with genome sequencing for optimal diagnostic yield. In the three remaining undiagnosed cases, extensive research is underway to identify potential genetic paradigms that have not been captured by current technologies. 

Our site regularly conducts exome sequencing, as needed, on infants suspected to have a genetic disease—while this clinical option may have been available to the majority of the patients had they not enrolled in the study, the results would not necessarily have been returned while the child was still admitted to the hospital. Our site also offers rapid exome sequencing on a clinical basis, though this is still considered a significant expense; cases require discussion with the institutions’ director of clinical laboratories. This study represents the first implementation of rapid genome sequencing at our site.

Our work shows the importance of close collaboration between clinical providers and research teams; ongoing research-based genomic analysis will hopefully provide invaluable insights into our non-diagnosed cohorts. Further research in one solved case has led to a new mechanistic understanding of the disease and enabled the phenotype expansion of a genetic disorder. Cascade testing for incidental familial variants has been performed and has identified several affected family members in family 1, allowing for a change in care and prospective monitoring for a potential fatal arrythmia disorder. We continue to explore potential causative variants in the yet-undiagnosed cases, establishing cell models and performing RNA and long-read DNA sequencing to identify novel mosaic, structural, or regulatory variants that may be associated with disease. Understanding the genomic etiologies that evade our current testing paradigms will strengthen the diagnostic capability of genomic sequencing as it becomes more commonplace in clinical care.

## Figures and Tables

**Figure 1 jpm-12-01924-f001:**
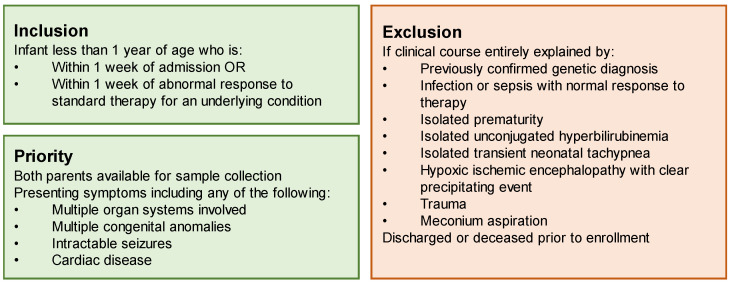
Inclusion and exclusion criteria, utilized specifically by the Duke site, adapted from the overarching Vermont Oxford Rady Children’s Genomic Network study’s inclusion and exclusion criteria.

**Figure 2 jpm-12-01924-f002:**
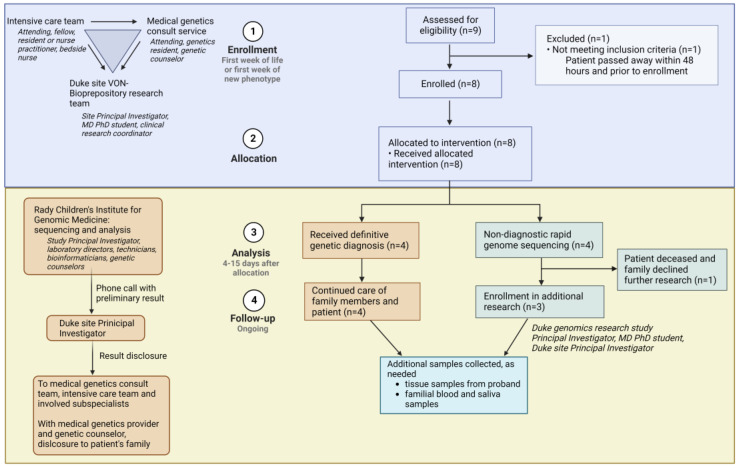
Study team and design flowchart showing overall study enrollment, as well as coordination, communication, and interactions among multiple medical providers and research teams across two institutions. Image created with Biorender.com accessed on 14 November 2022. VON: Vermont Oxford Network.

**Table 1 jpm-12-01924-t001:** Demographics and testing indications of enrolled cohort.

Category	Subjects
**Organ system involved**	
Hematologic	4/8
Brain	4/8
Cardiac	8/8
Skeletal	8/8
**Primary indication for testing**	
Multiple congenital anomalies	5/8
Coagulopathy/Inflammation	2/8
Metabolic aberrancy	1/8
**Sex**	
Female	4/8
Male	4/8
**Self-reported parental race/ethnicity ***	
White	11/16
Hispanic	2/16
Black	2/16
Native American	2/16
Asian	1/16

* One set of parents each reported multiple races/ethnicities: one parent reported Asian/White, and the other reported Hispanic/White.

**Table 2 jpm-12-01924-t002:** Detailed clinical summary and genetic testing results of all patients enrolled in the study. rWGS: rapid whole genome sequencing; FISH: fluorescence in situ hybridization.

Patient	Age at Enrollment	Time to rWGS Result	Clinical Features	Sequencing	Molecular Findings	Diagnosis	Diagnostic Test *		Incidental Findings	Change in Management	Deceased		Consented to Further Research
							rWGS	Other			Prior to Result	After Result	
1	3 days	11 days	Dandy-Walker malformation, eye abnormalities, congenital heart disease, sacral anomalies, coagulopathy	Trio	*CAPN15* compound heterozygosity [hg19] chr16:84116-692192, del(16p13.3); *CAPN15* (c.2594T>C, p.Leu865Pro) *PKP2* heterozygosity (c.235C>T, p.Arg79Ter)	Oculogastrointestinal neurodevelopmental disorder	X	microarray	X	familial cascade testing, cardiac surveillance	X		X
2	3 days	9 days	cleft lip, microcephaly, eye anomalies, brain anomalies, Dandy-Walker malformation, congenital heart disease, heterotaxy, skeletal anomalies, single umbilical artery, skin abnormalities	Trio	Mosaic terminal loss 13q [hg19] chr13:35,073,749-115,107,733	Mosaic monosomy 13		blood microarray and buccal FISH				X	X
3	7 days	14 days	schizencephaly, eye abnormalities, bone abnormalities, congenital heart disease	Duo (rerun as Trio)									X
4	7 days	13 days	Hemihypertrophy vs. hemiatrophy, colonic atresia, congenital heart disease, skin abnormalities, skeletal abnormalities, sacral anomalies, single umbilical artery, club feet	Trio									X
5	6 months	4 days	heart failure, congenital heart disease, coagulopathy, inflammatory abnormalities, subdural hemorrhage, femur fracture	Trio	*TTN* de novo heterozygosity *TTN* (c.5253del, p.Lys17508AsnfsTer17)	Titin-related disorders	X					X	
6	10 days	14 days	encephalopathy, intracranial hemorrhage, brain anomalies, congenital heart disease, hepatomegaly, hepatic failure, adrenal hyperplasia, coagulopathy, skeletal dysplasia	Trio	*GATA1* de novo heterozygosity- variant of uncertain significance *GATA1* (c.529G>A, p.Gly177Arg)	Suggestive X-linked hematologic condition; non-diagnostic given variant classification, female sex of patient, and inability to pursue further clinical testing after patient’s demise	X				X		
7	4 days	5 days	pectus excavatum, hypotonia, congenital heart disease, talipes equinovarus, respiratory insufficiency, cleft palate, club feet	Trio	*STAC3* homozygosity *STAC3* (c.851G>C, p.Trp284Ser)	*STAC3*-related congenital myopathy	X			anesthesia precautions for malignant hyperthermia risk			
8	1 day	15 days	congenital heart disease, lactic acidosis, intracranial hemorrhage, hyperglycemia, fractures	Trio									X

* Indicates the test(s) that resulted in conclusive molecular diagnosis.

## Data Availability

The data presented in this study are available on request from the corresponding author. The data are not publicly available due to patient confidentiality.

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
