# Peer review of "Rapid Whole Genome Sequencing in Critically Ill Neonates Enables Precision Medicine Pipeline"

_jpm, 2022, doi:10.3390/jpm12111924_

Round 1

Reviewer 1 Report

Given the small cohort, I felt the paper lacked any particularly novel conclusions. However, it it was sound overall. I have minor suggestions below.

Table 2:  It was initially not clear to me why Patients 2, 3, 4, and 8 did not have an “x” under rWGS, when all of these patients received rWGS. However, upon further review, I realize you are trying to specify “which test (if any) was it that was diagnostic in each of these cases?” I don’t think changes are needed, but it did make me pause.  If readers were presented with this table in isolation (ie, hadn’t read the rest of the paper), I think they would also pause.   Consider exploring ways to make this clearer?  I also wonder if you should specify in a footnote why some cells have a dash and others are blank.

Lines 167 – 176: appreciate this detail / discussion, as it highlights difficulties in the logistics of testing that I’m sure many hospital systems share.  Were there many patients that you wished to include in the study, but logistical challenges and inability to obtain parental samples or in-person consent prohibited inclusion?  Per Figure 1, “both parents available for sample collection” was a priority criterion, but not an exclusion criterion.  Could you specify whether this ended being a significant factor in which families were invited to participate?

Lines 209 – 211: “As with any genomic test, variants in genes without an established connection to disease will not be reported.”   I would re-word this sentence, as this depends on individual laboratory reporting practices.  I was recently involved in a case in which rapid genome was ordered (clinically, not on a research basis) and the lab included a table of variants in various genes that did not yet have definitive gene-disease associations yet in their report.   So I think your statement is (usually) true, but it varies by lab.

Line 215 – 224: I think this paragraph could be strengthened by adding a short discussion on how your stringent enrollment criteria may/may not alter diagnostic yield.  Given you had a limited number of free genomes to implement, it seems you tried to include patients who had the highest likelihood of having a genetic disorder (criteria highlighted in Figure 1).   Would these have been cases that the on-call team would have sent out exome or genome on clinically?  Does your institution have certain inclusion or exclusion policies for sending rapid testing on a clinical basis?  And if so, did that enable some of these patients to receive testing and thereby diagnosis? Or would this testing have occurred clinically either way?  

Reviewer 2 Report

This work is interesting primarily as a description of a pilot project for the examination of newborns in order to improve medical care.

I really liked the clear selection criteria, routing and design of the study.

Due to the small size of the group of subjects, none of the patients for whom the results of the genomic study were critical for treatment did not get into it.

I would ask the authors to add to Figure 2 the timing of all stages of the study.

In addition, an error has crept into table 1 in the ancestry section. It is necessary to check the number of patients of different origins, taking into account the fact that there are 8 of them in total.

It is not very clear how the family analysis allowed to increase the pathogenicity of compound heterozygous variants in the CAPN15 gene? Have there already been sick children with a similar phenotype in the family who have not been identified the cause of the disease? I think it is necessary to describe these results in more detail and convincingly.

In general, the work deserves publication after the elimination of minor problems.
